# Mobile Applications Accessibility: An Evaluation of the Local Portuguese Press

Tatiana Santos Gonçalves [1,2,*], Begoña Ivars-Nicolás [1] and Francisco Julián Martínez-Cano [1]

1   Department of Social and Human Sciences, University Miguel Hernández, 03202 Elche, Spain; bivars@umh.es (B.I.-N.); francisco.martinezc@umh.es (F.J.M.-C.)
2   Centre for Studies in Education, Technology and Health, Polytechnic Institute of Viseu, 3504-501 Viseu, Portugal
*   Correspondence: tsantos@umh.es

**Abstract:** The local press has always played a central role in the Portuguese society. Recently, new innovative technological projects to develop mobile applications and focus on local journalism in Portugal have emerged. These initiatives allow the development of better and more appealing services for local users. However, due to the important social role of the local press, this also brings along some responsibilities. Our main research goal is to study the accessibility issues in local journalism in Portugal. To this end, we first describe the current situation of local journalism in Portugal and some accessibility issues raised by the appearance of mobile applications. We then develop a simple checklist that allows the assessment of whether these applications have prevented social exclusion and facilitated the access of local information to a wide range of users, including disabled citizens. This tool provides the regional news publisher with information to improve its democratization of access to local information in Portugal. Using the cognitive walkthrough method, we illustrate the proposed framework by presenting case studies of five mobile applications in Portuguese local and regional press. This study concludes that despite the great potential that mobile applications showcase, several accessibility issues have not been properly addressed.

**Keywords:** accessibility; mobile apps; local journalism

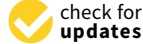



## 1. Introduction

The local press in Portugal has endured several adversities in the last decade. The few advertising revenues, as well as the lack of sales and own resources, will inevitably force many publications to close their doors. This situation is due, among many other reasons, to the poor utilization of the potential of the Internet and the new technologies by local newspapers in Portugal. In particular, the use of mobile applications has completely changed the way in which the sector of local media operates and the Portuguese local media has not adapted fast enough to this new context. However, in recent years, several cutting-edge projects, backed mainly by large technology corporations such as Google, have appeared in Portugal to develop mobile applications focusing on local and regional journalism. These new initiatives have brought promising opportunities to counteract the decline of the sector.

The advantages of using mobile platforms are clear: they present suitable features for sharing information and they are modular, portable and distributable. However, the development of the Portuguese local media comes with certain responsibilities as they play a central role in the local community, by making available forums for public discussion and preserving the identity and cultural ties of the region [1]. Furthermore, when it comes to accessibility, new applications are often available before we can assess if they are suitable in terms of accessibility, an aspect of technology that has the potential to create inequalities among users. For a better and more democratic dissemination of regional news, it is important that local news is produced in such a way that it is accessible to the greatest possible number of citizens regardless of their social status, age, or physical and mental

conditions [2]. This effort towards equality should be even greater when it comes to the use of technology, as elderly or people affected by disabilities or sociodemographic limitations face numerous accessibility challenges [3–6].

The main purpose of this work is to analyze accessibility issues raised by the emergence of mobile applications for local journalism in Portugal. We study how the irruption of these new platforms and technologies brought up not only new possibilities to this sector, but also the responsibility to prevent social exclusion and to facilitate access to elderly or disabled people. In order to achieve the main goal of this work, which is based on the cultural and socioeconomic nuances of the different Portuguese regions, we develop a simple guideline for the assessment of accessibility of mobile applications for local journalism in Portugal. This guideline is constituted by a set of indicators (checklist) for proper accessibility practices. This set of indicators is presented as a practical tool to help design mobile applications for Portuguese regional and local newspapers with adequate accessibility features. To illustrate the proposed framework, the manual accessibility audit was performed by the authors of this paper using the cognitive walkthrough method of five relevant mobile applications, namely Diário do Minho, Açoriano Oriental, Dnotícias, Folha do Domingo and Repórter no Mundo, to determine to what extent the accessibility criteria proposed has been met by these innovative applications.

### 1.1. Portuguese Regional and Local Press

Regional journalism refers to the coverage of events and topics on a small and local scale that would usually not be followed by the larger mainstream media, which tends to cover stories of interest to a wider national audience. In that sense, regional journalism has always been an essential part of local life, where local journalists report events from everyday life, propose local debates, foster democracy and keep an eye on those in positions of power. For all these reasons, it is necessary to ensure that this information is not only published, but that it is transparent and accessible to the plural citizenship of the local audience in which we live [2]. The local media has not only provided local news but has also helped people imagine themselves as part of a community, connecting them by more than physical proximity [7].

However, local journalism is undergoing a revolution, driven in large part by the rise of digital and, more particularly, by the use of mobiles devices, which have changed the ways in which users access, find and share information [8]. This new context challenges the traditional business models and journalistic practices that have persisted in the last decades in the regional news media [9,10]. Most developed countries share certain commonalities with respect to these changes, such as the decline of newspapers' print circulation, but each community, region or nation has adapted differently to this new context and needs to be analyzed independently. For example, the adaptation to digital formats in countries such as Norway is very different than in Portugal. Data shows that most of the Norwegian population is familiarized with the use of the Internet, while in countries such as Portugal there is still about 30% of the population that are non-internet users [11]. This fact has straightforward implications in the ways in which local newspapers are consumed and, therefore, must be considered when studying accessibility issues. This study considers this issue at a national level and focuses on the Portuguese case.

The number of regional and local newspapers that are closing their doors in Portugal has been increasing in the last decade [12]. One of the main reasons for this decline is due to the fact that the Portuguese local media has always been a subsidized service and, therefore, suffered deeply from the end of the local press support policies in Portugal in recent years. Without this support, the sector was left with no alternative distribution network and low levels of professionalism among local journalists. Moreover, the aging of the Portuguese population, especially in the countryside, and the challenges posed to the elderly by the use of technological devices, can be pointed out as key factors that accentuate the problem of digitalization of regional publications in Portugal [13,14]. This

decline in the sector forced the revision of the longstanding strategies and the need to seek new approaches to become sustainable.

One can distinguish several possible issues when trying to incorporate the use of emerging technologies (such as mobile applications) by the regional and local press: low levels of professionalism in the Portuguese staff (many local journalists are not professionals or sufficiently qualified to use new technologies) together with the lack of attention to incorporate modern technology resources and indifference in the face of cultural, economic and technological changes [14].

Although the falling rate of Portuguese local newspaper subscriptions and their revenue from advertisement is likely to continue decreasing in the coming years, recently novel technological initiatives have been carried out to address these problems and promote the Portuguese local press. In this work, we will describe and analyze five of the most representative Portuguese mobile applications, some of which are funded by the Google Digital News Initiative (DNI).

The regional media has historically played a very important role in journalism in Portugal, averaging around 50% of the total national share market in the country [15]. The Portuguese local press is well known for its sense of proximity, which reflects the very closed relationship between the readers and the news organization's region [13]. The closer the media is to its audience, the easier it will be to strengthen ties and bolster participative democracy [16,17] for other aspects that influence participation in the Portuguese media. According to [18], this proximity is seen by the users as a platform to express and convey local community problems and preserve the culture, the customs and the local habits. Portuguese local journalism plays a fundamental role not only in the region or place in which it is inserted but in the type of information it disseminates, maintaining strong ties between locals and the numerous Portuguese immigrants scattered around the world. With a national press that is not so sensitive to this type of content, the local press may become the only means of information for immigrants. A similar situation faces elderly people or citizens living in low-density regions, particularly in the countryside areas of Portugal, that have in the Portuguese regional press a unique tool for its valuation and to preserve their identity, cultural and historical ties.

### 1.2. Accessibility of Mobile Applications in the Regional Press

Accessibility is a major concern in today's society as elderly and disabled people are at risk of being excluded from the use of information technologies. In this sense, it is paradoxical that the change from print to internet content has generated barriers in terms of access to local information for certain groups of users. The current EU accessibility of mobile applications legislation can be found in [19]. Accessibility has been defined in various ways but here we will adopt the standard definition where the accessibility is defined as the usability of a product, service, environment or facility by people with the widest range of capabilities. This means that all users can perceive, understand, navigate, and interact with it [20].

Our research aims to provide the regional news publisher with information to improve its democratization of access to information. To achieve this goal, it is necessary to design mobile applications considering accessibility issues so as to reduce potential barriers. We believe that, for a better and more democratic dissemination of regional news, it is important that local and regional news can reach the largest number of citizens. In this regard, we start from the premise that regional press outlets are essential sources of information and believe that an improvement in the accessibility of these platforms would stimulate the dissemination of local public affairs in a more balanced and democratic way among all citizens. This effort towards equality should be even greater when it comes to the use of technology, as elderly or people affected by a disability or sociodemographic limitation face numerous accessibility challenges [3]. Most disadvantaged citizens in rural areas in Portugal are, firstly, elderly, followed by immigrants, foreigners, non-experienced

and those with few economic resources. Hence, these groups must be taken into account when developing new platforms to communicate via mobile devices.

In fact, there is a growing elderly population that suffers from age-related disabilities and new devices are developed before the accessibility issues for the previous models have been assessed. There are mainly four requirements that mobile communication systems for disabled and older people should meet: personal communication (to help the communication of readers with restricted movement), security (to assists in situations of accidents or illness), social integration (to enhance social inclusion and access to education) and autonomy (to provide more opportunities to carry out an independent way of life).

Most of the documents that propose rules and standards for web accessibility are based on the Web Content Accessibility Guidelines 2.1 (WCAG 2.1) which is developed through the World Wide Web Consortium (W3C) process in cooperation with individuals and organizations around the world [21]. Requirements include, among others, providing equivalent alternatives to visual content, and providing context and guidance information to help users with special needs to understand pages and clearly providing consistent and clear navigation mechanisms [21]. A characterization of the current scenario of mobile application development considering accessibility issues for elderly people was presented in [22]. Portugal created the INCNESI National Initiative for Citizens with Special Needs in the Information Society (INCNESI, Iniciativa Nacional para os Cidadãos com Necessidades Especiais na Sociedade da Informação). INCNESI's main aims are to ensure that Portuguese citizens requiring special consideration are not excluded from the benefits of the information society. Based on these premises, several studies have emerged that analyze the accessibility of digital platforms. For example, the studies of the Rehabilitation and Accessibility Engineering Centre (Centro de Engenharia de Reabilitação e Acessibilidade) and the University of Trás-os-Montes and Alto Douro, CERTIC—UTAD [23,24], which focused on the evaluation of web accessibility of the contents of Portuguese municipalities. Indeed, CERTIC—UTAD [23] is included in the Portuguese government portal as a reference and accessibility guide for Portuguese web platforms. Thanks to the regional press, the small communities have access to local public information, but it is not enough for the information to be published, even if it is accessible and transparent to the majority; it must also serve a plural citizenship and cover as many social classes as possible even if the news organization is not 100% public [2,25].

The Android operating system is clearly the leader in the mobile phone industry in Portugal with 78% of the market share [26], see also [27]. The giant corporation Google Inc. has also presented a guideline for developers using Android that consists of the following set of best practices to address accessibility issues: (1) add descriptive text to user interface controls; (2) ensure that all the functionalities that can accept input (such as touches or typing) can be also performed with a directional controller, such as a navigation, D-pad (either physical or virtual) or trackball; (3) check that audio elements are shown together with a visual notification, to help deaf users; (4) turn on TalkBack and Explore by Touch, the two assistive technologies available for all Android mobile phones, and then try to interact with the application using only directional controls.

It is also worth mentioning the work of [28], where a usability guideline for web design, grouped in 11 different categories, was presented. Although some of the topics addressed were exclusively devoted to web pages, and therefore outside the scope of this work, there were six dimensions that could be applied to the mobile context, namely: target design (e.g., the older adult should not be expected to double click), use of graphics (e.g., icons should be simple and meaningful), browser window features (e.g., provide only one open window avoiding multiple overlapping), content layout design (e.g., avoid irrelevant information on the screen), user cognitive design (e.g., allow ample time to read information) and use of color and background (proper use of colors).

In order to assess the accessibility concerns of the elderly population we shall start from the set of guidelines for mobile applications given in [29,30], which can be divided into three dimensions: Perceivable (proper use of colors, contrasts and the existence of

non-text content as images, videos and sounds), operable (easy to navigate with content in logical sequence, navigation and terminology used should be simple, clear and consistent) and understandable (language should be simple and clear and important information should be highlighted, avoiding multiple stacked windows or blinking or moving content).

The requirements of mobile communication systems to assist disabled and elderly people can be succinctly summarized in four dimensions: (1) personal communication: for users with restricted movement, mobile technology enhances their chances of personal communication; (2) security: situations of illness, home accidents and so on, require a quick communication channel; (3) access to education and labor market: services such as remote work and remote learning contributes to social inclusion and autonomy of user with disabilities and (4) autonomy: the combination of personal communication, security and access to integrative services grants more opportunities for people with disabilities and older people to carry out an independent way of life [31].

Lastly, it is important to notice that there exist automated evaluation tools that can assist in the assessment of accessibility. However, as pointed out in [31], they can only check a limited number of aspects automatically and human judgment is required. Therefore, accessibility evaluation tools cannot determine accessibility; they can only assist in doing so.

## 2. Materials and Methods

After having analyzed the current situation and particularities of the Portuguese local press, we now aim at deriving a guideline based on the theoretical background described above that consists of a set of indicators. These will help the design of mobile applications for Portuguese regional/local newspapers and will be presented in such a way that designers and local journalists can easily verify whether the basic accessibility criteria are being met or not. Hence, the indicators are presented to be practical guidelines by means of a simple checklist to be kept in mind by design developers to achieve accessibility in mobile interfaces. The objectives of this work are threefold:

1. Design a guideline for the assessment of accessibility of new mobile applications of the local and regional press in Portugal;
2. Validate the proposed set of indicators by analyzing five of the most representative mobile applications in Portugal;
3. Identify possible accessibility issues and challenges of the emerging mobile applications of the local press in Portugal.

We have created a framework with two general dimensions of evaluation: design and content. Within each general dimension of evaluation, we derive several indicators to evaluate each dimension individually (Tables 1 and 2, below). We derived our set of indicators by identifying the key factors of the theoretical framework developed above and adapting the ones presented in [29,30] to the Portuguese context. The more standard indicators were directly derived from [29]. As the elderly population is an important part of the Portuguese local press users, some extra indicators were developed using [30].

After, we will analyze five case studies using cognitive walkthrough for accessibility evaluation for the following Portuguese mobile applications: Diário do Minho, Açoriano Oriental, Dnotícias, Folha do Domingo and Repórter no Mundo. These apps were selected for several reasons: they belong to the oldest local newspapers in Portugal, have more updated news and show a faster growth rate in the number of visits than other regional press outlets [32].

For the sake of ease and clarity, we propose the use of a checklist where the indicators are designed in such a way that the answers are positive when the information is available and negative when the information is not present, incomplete or inadequate. There exist several evaluation methods and they can be formal, automatic, empirical or informal. Here we focus on informal methods that include the so-called inspection methods. These provide a good compromise between the cost and implementation time on the one hand and the results they make it possible to obtain on the other. We have opted for this methodological

option as these indicators provide an easy tool for the evaluation for Portuguese local journalists that are, in many cases, not professionals. Amongst the inspection methods, we chose the cognitive walkthrough method that enables easy identification of a certain number of accessibility issues [33,34]. Hence, the authors propose the so-called checklist and expert accessibility audit performed by the authors during the cognitive walkthrough. Even if the assessment is subjective, both the checklist and the way in which they are evaluated allow transparency and verification and are derived from the analysis exposed in the theoretical framework. In the context of Portuguese local journalism, where journalists are often amateurs, it appears as a reasonable alternative to, for instance, Likert-type or open questions that require the analysis of a higher volume of information.

**Table 1.** Indicators for the analysis of the design of Portuguese apps.

| | Indicators of Evaluation of Design |
|---|---|
| 1 | The main page presents search tools with the icon of a magnifying glass or via the words "go" or "search". |
| 2 | The accessibility symbol is located and is visible. |
| 3 | The page has sufficient contrast between the background, the screen and the contents. |
| 4 | There are no flashing screen situations with intermittent visual effects. |
| 5 | New windows do not open in the application without user permission (such as advertising, videos, etc.). |
| 6 | The elements of the page are visually evident (such as buttons, icons, links, text and images). |
| 7 | The app does not exclusively use colors to differentiate the elements on the pages. |
| 8 | The pages have the same structure and positioning of elements. |
| 9 | The layout design is easy to navigate with consistency between the order of focus (guide) and the content in a logical sequence. |
| 10 | The user is informed about the location of the page. |
| 11 | The app identifies and describes possible data entry errors in forms. |
| 12 | There is contact information with the page editor via phone or email (public or personal). |

**Table 2.** Indicators for the analysis of the content of Portuguese apps.

| | Indicators of Evaluation of Content |
|---|---|
| 13 | Language is simple and clear. Important information is highlighted. |
| 14 | The main language of the page is identifiable. Offers the possibility to choose the language on the page and informs the user about language changes. |
| 15 | Facilitates audio alternative to texts and images. |
| 16 | There are text alternative such as images, sound or video. It presents different texts and audios for non-text objects (images, sound, video and graphics). |
| 17 | Guarantees the user's control over changes in content. |
| 18 | There is no automatic content update (e.g., Flashing, moving or scrolling). |
| 19 | Use of elements such as color, bold or italics to highlight part of the information. |
| 20 | Links to news, information or content on this or other platform are different from the rest of the text. |

The analysis begins with the app's homepage and a full navigation is continued both along and across the different pages. The experiments were conducted with a Xiaomi Redmi Note 8 smartphone running Android 5.1. The data collection and the testing of the apps were performed by the researcher of this paper from March to June 2021. In the following subsection, we will describe the apps that have been considered and analyzed in this work.

### 2.1. Mobile Applications for Local Press in Portugal

2.1.1. Diário Do MINHO: An App with a Focus on the Region of Minho

Diário do Minho is a daily information newspaper focused on the northern region of Minho and was founded on 15 April 1919. The journal launched its mobile app in August 2018, and it is available for both Android and iOS platforms.

We selected the mobile application Diário do Minho because it is one of the most representative in the north of Portugal. According to the audience survey conducted by Bareme Imprensa Regional [31], the average audience of this newspaper reaches around

70,000 daily readers. In addition to the full edition of the Diário do Minho newspaper, this application offers specific content for mobile devices. The platform presents local news and reports, specifically about the city of Braga, as well as local and national sports.

### 2.1.2. Açoriano Oriental: The Oldest Daily Portuguese Newspaper

Açoriano Oriental was founded 18 April 1835 and is the oldest daily Portuguese newspaper. It is one of the ten oldest in the world that has continuously published on a daily basis. The newspaper is focused on regional information from the Azores archipelago and maintains an editorial line of freedom, rigor and exemption, having as its main goal "the free administration of the Azores by the Azoreans" in the defense of a wide political and administrative autonomy.

Its mobile application was created in 2017, bearing the same name as the newspaper. Available for Android and iOS, the app features the print edition of the newspaper in a digital format (PDF). To access this content, the reader must sign up with the app. In addition, the application also provides the online version of the Azores' magazine. The app provides daily news and reports, with local and regional content about the Azores archipelago.

### 2.1.3. Dnotícias Pt: An App from a Centenary Madeiran Newspaper

The newspaper Diário de Notícias da Madeira is a centenary daily newspaper from the archipelago of Madeira, with an average circulation of 10,854 copies. It is also the Portuguese regional newspaper with the greatest expansion and circulation and has around 5600 subscribers [32]. It was founded in 1876 and in 2010 was awarded with "European Newspaper of the Year" in the 12th edition of the "European Newspaper Award", in the category of local newspaper, an unprecedented distinction in the history of Madeira journalism.

Its web version is the largest Portuguese regional daily newspaper, with great expansion among the immigrant communities in South Africa, Venezuela, Brazil and the United States. Its app was created in 2016, the date of the newspaper's 140th anniversary, and bears the same name as the portal: Dnotícias.

### 2.1.4. Folha Do Domingo: A Newspaper from the Algarve Diocese

The Folha do Domingo newspaper is a newspaper created in 1914 by the Algarve diocese. It is the second oldest newspaper in this region and one of the oldest in Portugal. It is one of 31 publications with at least a century in this country [32].

Its mobile app was created in 2020 and is available in Android and iOS versions. This application does not have a digital version of the newspaper, betting on content created exclusively for the mobile application. Regarding the content, the application provides news about the Algarve diocese and also general news about the cities that are part of this region, with sections on politics, education, culture, society and sports. The platform has a tool for accessibility, identified by the accessibility symbol. By clicking on this symbol, a menu is available with resources for people with disabilities to access the content. To improve the understanding of the content available on this platform, the user can choose to obtain contrast between text, images and screen, highlight links, increase the size and spacing of the text or automatically stop the animations present in the app.

### 2.1.5. Repórter No Mundo: An App for Local News in the Central Region of Portugal

Regarding the consumption of regional and local journals, the central region of Portugal has a strong implementation of this type of press. In these areas there are higher than average audiences, with 73.4% of news consumption being regional/local newspapers [32]. This may be due to the fact that historically this district has presented a large number of publications of a regional nature.

The application selected for analysis stands out as one of the most representative of the Portuguese central region mobile applications. This mobile application was created in

2018 by the regional media news Região de Leiria in collaboration with the Polytechnic Institute of Leiria and funded by the Google Digital News Initiative (within the European programme Digital News Innovation Fund). Região de Leiria has been a pioneer in the transition to online news among regional journals in Portugal and has the biggest update of contents and growth of visits and page views. The application was created as a means to foster the local economy and cultural ties as well as for broadcasting local news. The app has a particular interest in enhancing a tight relationship between users of the region who are abroad and those who are in Portugal and is currently available for both Android and iOS platforms.

### 3. Results and Discussion

In this section we present the results obtained from our analysis of the apps together with a discussion of their importance in the Portuguese context. The section is divided in two parts: the design and the content of the apps. The results of the analysis showed that the application Folha do Domingo meets the highest number of accessibility indicators proposed in this study, with 18 criteria met. On the other hand, Diário do Minho presented the lowest number of positive indicators, with only 11 criteria fulfilled.

*3.1. Results of the Analysis Using the Proposed Set of Indicators for the Design of Apps*

According to the data obtained in the first dimension, design (Table 3 below), the apps Repórter no Mundo and Folha do Domingo display in the main page a search tool with a magnifying glass icon, complying with criterion (1), while the remaining apps do not present this type of tool on the first page, only on the pages that follow. Dnotícias, for example, presents the search tool via the word search, but this tool is not visible in the application. The lack of this type of tool or the difficulty in finding it directly affects citizens with advanced age, little web experience or mental disabilities [29,30]. Regarding indicator (2), the Folha do Domingo app was the only platform to display an accessibility symbol. The other platforms did not present any symbol or other type of tool for this purpose. According to [29] the lack of an accessibility symbol or any link for this purpose directly affects citizens with physical or mental disabilities. This type of resource informs this group of users that this application offers a set of tools that allow people with disabilities to use the resources that this platform provides.

**Table 3.** Results of the evaluation of the *design* of Portuguese apps.

| | 1. Main Page Presents a Search Tool | 2. The Accessibility Symbol Is Present and Visible | 3. The Page Has Sufficient Contrast | 4. No Flashing Screen Situations | 5. Does Not Open New Windows without Permission | 6. The Elements of the Page Are Visually Evident |
|---|---|---|---|---|---|---|
| Diário do Minho | No | No | Yes | No | No | Yes |
| Açoriano Oriental | No | No | Yes | No | Yes | Yes |
| Dnotícias | No | No | Yes | Yes | Yes | Yes |
| Folha do Domingo | Yes | Yes | Yes | Yes | Yes | Yes |
| Repórter no Mundo | Yes | No | Yes | Yes | Yes | Yes |

Even with the lack of the accessibility symbol in most apps, all platforms presented a sufficient contrast between the background, screen and contents, fulfilling criterion (3) of this study. In fact, the color contrast between the background and the text are recommended for people with low vision to view the screen. This type of resource facilitates the reading by users with a loss of perception or sensitivity to contrast due to aging [28]. The newspapers Diário do Minho and Açoriano Oriental did not meet indicator (4) of this work. Both platforms present issues with flashing screens and too many visual effects. In a mobile application or on a website, intermittent or sparkling visual effects on the screen should not be used. This type of effect would directly affect people with photosensitive disabilities,

making it difficult to read the text and causing health problems [21,30]. Diário do Minho failed also to comply with criterion (5). During our analysis, we noted the appearance of advertising pop-up windows without the explicit permission of the user, thus losing control of the display of these windows on the screen. According to W3C [29,30] when opening new windows on the screen without the user's permission (such as photos, videos and advertising), citizens with blindness, low vision, with restricted field of vision or with cognitive impairment do not realize that the active window is new, becoming disoriented. When trying to close these new windows, the user can close the pages without realizing the browser instance and exit the application. In this case, the recommendation of the W3C [29] is that access to pages or platform features should be the user's choice. Likewise, the applications comply with indicator (6), as the elements of the pages are visually evident, facilitating navigation and reading of different groups of users with disabilities such as citizens with low vision, cognitive impairment, advanced age, mental disabilities or with little web experience.

While conducting our analysis we observed that all the applications comply satisfactorily with the accessibility indicators (7) and (8) proposed in this dimension (Table 4 below). The elements of the pages are visually evident, facilitating navigation and reading for different groups of users with disabilities. Besides the use of colors, other features are used to differentiate the elements present on the pages such as links, icons, buttons, texts and images. This eases the navigation for people with low vision or color blindness, since this type of user does not perceive color difference and may confuse or misunderstand information [30]. The apps also have the same structure and positioning of the elements throughout the different pages. We noted, however, that the Diário do Minho app was the only one that did not fulfill indicator (9) of the criteria, with the absence of logical sequences of the content. It is important to highlight that all these accessibility features guide the navigation of groups of users with physical or mental disabilities, with advanced age and without web experience, allowing them to scroll through a set of pages and to know where they are at every moment [29].

**Table 4.** Results of the evaluation of the *design* of Portuguese apps.

| | 7. Does Not Exclusively Use Colors to Differentiate the Elements on the Pages | 8. The Pages Have the Same Structure | 9. Layout Design Is Easy to Navigate | 10. Identifies and Describes Possible Errors in Forms | 11. Informs the User about the Location of the Page | 12. Contact Infomation |
|---|---|---|---|---|---|---|
| Diário do Minho | Yes | Yes | No | Yes | Yes | Yes |
| Açoriano Oriental | Yes | Yes | Yes | Yes | Yes | Yes |
| Dnotícias | Yes | Yes | Yes | Yes | Yes | Yes |
| Folha do Domingo | Yes | Yes | Yes | Yes | Yes | Yes |
| Repórter no Mundo | Yes | Yes | Yes | Yes | Yes | No |

All platforms obtained a positive evaluation on indicator (10), informing the users about its location during navigation. As recommended by W3C [29], the applications present navigation links through a hierarchical list, guiding the reader from the main page to the page they are on, passing from one element to another and navigating in series. Likewise, all the platforms properly assist users in filling out forms, identifying and describing possible data entry errors, as described in criterion (11). Each error is automatically identified and described to the user through text. After the correction, a confirmation screen appears on the page stating that the form can be sent. The app Repórter no Mundo was the only one that failed to comply with indicator (12) in our guidelines, without presenting contact information for the page editor via telephone or email. We consider that the non-compliance with these accessibility indicators is a serious flaw in this application. In addition, the lack of contact information for the application editor directly

affects senior citizens and those with little experience in using ICT. During data collection, the application presented problems with user registration, as well as with uploading videos and photos for composing news. Faced with these types of problems, the presence of a support area in the application with information about email and phone is essential to assist an elderly user or the reader who understands little about the use of web tools. In addition, in Portugal, a large range of the population is over 65 years old and actively consumes news in newspapers and on the web [35]. Therefore, it is necessary to present a support section in the menu of this application to provide this type of tool, so as to meet the needs of this type of audience and also of other groups of users.

*3.2. Results of the Analysis Using the Proposed Set of Indicators for the Content of Apps*

In the second dimension, content (Table 5 below), the applications Repórter no Mundo, Diário do Minho, Açoriano Oriental, Dnotícias and Folha do Domingo achieved a positive evaluation of indicator (13). They presented simple language with short sentences and without expressions that are difficult to understand. For [30,31], this type of resource facilitates the reading and textual understanding of people with advanced age or cognitive impairment since these users have difficulty processing information. Deaf users (whose native language is sign language) still have difficulty processing written language in a complex way. These authors warn that the text in mobile applications must be easy to read and understand, not requiring a higher level of instruction than elementary school. When the text requires a more advanced reading ability, complementary information should be made available to explain what illustrates the main content. These apps present such complementary resources by means of additional clarifying pictures or underlining the important text and therefore complying with the requirements for this indicator.

**Table 5.** Results of the evaluation of the *content* of Portuguese apps.

| | 13. Language Is Simple and Clear | 14. Main Language of the Page Is Identifiable | 15. Facilitates Audio Alternative | 16. Text Alternative Such as Images, Sound or Video | 17. Guarantees User's Control | 18. There Is No Automatic Update of the Content | 19. Use of Elements Such as Color, Bold or Italics | 20. Links Are Different from the Rest of the Text |
|---|---|---|---|---|---|---|---|---|
| Diário do Minho | Yes | No | No | No | No | No | Yes | Yes |
| Açoriano Oriental | Yes | No | No | No | Yes | Yes | Yes | Yes |
| Dnotícias | Yes | No | No | No | Yes | Yes | Yes | Yes |
| Yes | Yes | Yes | No | No | Yes | Yes | Yes | Yes | Yes |
| Repórter no Mundo | Yes | No | No | No | Yes | Yes | Yes | Yes |

However, we verified that there are important issues that are not properly addressed, thus compromising the accessibility to information of groups of users with limitations. According to the results, the following accessibility indicators were not met by the apps proposed: (14) the main language of the page is identifiable and offers the possibility to choose the language on the page and informs the language change and (15) facilitates an audio alternative to texts and images. We noted that only the app Folha do Domingo obtained a positive assessment for criterion (16) as it presented an alternative text and audio for non-text objects (images, sound, video and graphics). The lack of identification of the main language and the choice of language in the page mainly affects the navigation and the access for foreign citizens to the application content. This type of failure shows that the applications were not developed with this group of users in mind. Another problem detected during our analysis was the lack of sound or textual alternatives for non-text objects such as images, audio, videos or graphics presented in most of the apps (as mentioned before, only the app Folha do Domingo presented this type of accessibility feature). We consider it essential to have a sound or textual alternative for videos that do not include audio tracks. In the case of videos that contain audio, subtitles could be offered.

In addition to the subtitles the availability of an audio description feature is desired, that is, a clear and objective description, through audio and text, of all the information presented in photographs, podcasts or graphics [21,30]. These resources are essential for people with hearing, visual or cognitive impairments, and advanced age, as these groups may not have access to important information due to the lack of these types of tools on these platforms.

The application Diário do Minho did not meet indicator (17), as it did not guarantee the user's control over changes in content, or indicator (18) as it presented automatic content updates with refresh or pages moving automatically in the content. We noticed that when reading news and reports presented in this app, these automatic updates removed the user's autonomy of choice, ending up confusing and disorienting users with advanced age, less web experience and with mental and physical disabilities.

Finally, our analysis showed that, in addition to colors, all the applications use other types of elements to highlight information (19) and in most cases link the text to news, additional information or content that is different from the rest of the text (20). Regarding the first criterion, the applications mainly use icons representing actions or information to highlight the information. We observed that the user is informed about an error or an action is indicated to the user through a message accompanied by an icon and not just by a text written in a different color. This type of resource mainly serves blind or color-blind users who perceive colors differently or are not able to perceive them and, therefore, may misunderstand information and make navigation errors. The apps apply the same feature for using links in content. We remark that the links appear differently from the rest of the content or outside the body of the text, serving mainly users with low vision or cognitive disabilities.

## 4. Conclusions

In this paper, we have reviewed the difficult current situation of the regional press in Portugal and discussed how the new initiatives towards the increasing digitalization of the regional publications can bring opportunities and also social challenges, such as the one of accessibility. We have proposed a guideline to help the design of mobile applications of regional news so that the most relevant accessibility issues can be assessed. From this study a checklist with a set of 20 indicators has been derived. We have analyzed and evaluated five of the few and most advanced web applications of the regional press in Portugal by means of the proposed criteria. Despite the increasing interest of regional journals to develop mobile applications, and based on the analysis carried out, this work has shown that there are still significant accessibility issues that the developers of these applications need to address. Our study confirms that the deficiencies located in previous studies, e.g., [4,30,31], continue to exist in these mobile applications and, therefore, they do not comply with some of the accessibility requirements to provide support for other special needs of the elderly and people in disadvantaged social groups. In the same way, we can corroborate that some aspects of the configuration of the systems are not attentive enough to the elements related to their accessibility. Hence, there are requirements that should be addressed early on in the design phase in the development of the application. The regional press must continue to improve in the conception of methods to increase the accessibility of mobile applications and this paper provides new insights and tools to developers and journalists to strive for that goal.

**Author Contributions:** Conceptualization, T.S.G., B.I.-N. and F.J.M.-C.; Formal analysis, T.S.G.; Investigation, B.I.-N.; Methodology, T.S.G.; Visualization, F.J.M.-C.; Writing—original draft, T.S.G.; Writing—review & editing, B.I.-N. and F.J.M.-C. All authors have read and agreed to the published version of the manuscript.

**Funding:** This research received no external funding.

**Institutional Review Board Statement:** Not applicable.

**Informed Consent Statement:** Not applicable.

**Data Availability Statement:** Data sharing not applicable.

**Conflicts of Interest:** The authors declare no conflict of interest.

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
