# Peer review of "Mobile Applications Accessibility: An Evaluation of the Local Portuguese Press"

_informatics, doi:10.3390/informatics8030052_

Round 1

Reviewer 1 Report

Title (informatics-1309411): Mobile applications accessibility in the local press in Portugal

The paper offers a relatively simplistic analysis of the usability of selected mobile applications. I believe the manuscript fails to meet the criteria for research papers. Therefore, I do not recommend publishing it in Informatics. Please find my arguments below:

The manuscript seems to lack features of an informatics paper. It resembles a social sciences paper. I believe the content, materials, and methods, followed by results and conclusions, to be presented in a manner typical of social sciences, not engineering sciences. I learnt much more about journalism in Portugal than about mobile applications and issues related to their deployment and use.

The abstract is too general. I suggest revising the abstract so that it is more a cross-section of the manuscript, better structured, and more specific.

The literature review is hardly sufficient. The authors failed to demonstrate a research gap. What is the goal of the manuscript? What are the results of similar research to date? What has been investigated so far? Have the authors proposed a research hypothesis? A research question?

Why should a local newspaper publisher invest a substantial amount in a mobile application? What are the differences between a mobile application and an RWD or AWD (or AMP) website? Lastly, what benefits can a mobile application offer to the user so they would install it? Why should I install additional software (a mobile application) when I can access all I need using a mobile website? These are the key questions for publishers and users. The paper did not offer answers.

I find the research superficial, and the issue of mobile application use and investment significantly oversimplified. Application's usability and accessibility is tested during design and prototype use. It is at this stage that the decision to invest in an application is made. At this point, the investor already has a business profile for the operations, knows the audience profile, and should know its strategic goals (should know what target conversion means in their particular case). I am under the impression that the manuscript fails to touch upon these economic issues of application deployment, which are important for mobile applications.

Will a mobile application solve the problems of regional publishers? What would motivate the reader to install such an application? Why would I install it when articles are available on a mobile website? Why would a publisher invest in a mobile application when articles are available on a mobile website?

I have a feeling the paper presents mobile applications as a panacea for problems of local newspapers in Portugal. Still, this point is not supported by the literature review. Can mobile applications resolve the issue of dwindling numbers of subscribers? The literature review offers no answer to this.

Do the elderly in Portugal commonly use mobile applications? If so, provide research results to confirm it, please. Personally, I doubt it. I think it is the other way round. It is the traditional (such as print) media that cater to the elderly, while mobile applications and new technologies are targeted at younger generations. The authors employed a completely reversed research model. It appears that the Portuguese newspaper industry focuses on the elderly. Is it so?

Section 1.2 touches upon too many topics and does so superficially, at that. First, the reader learns about the ageing population of Portugal. Next, the authors refer to the W3C and INCNESI, but completely disregard the WCAG(!). ‘All of a sudden’, Android comes in only for the piece to end with a short presentation of so-called mobile communication systems requirements. This part lacks order and causality.

Who collected the data? Who tested the applications? What was the size of the test group? Was cognitive walkthrough employed? Many methodological questions arise.

Where are the results discussed? The manuscript does not contain a discussion of the present results (authors' observations) or a discussion in the context of international literature. It is a serious shortcoming.

COVID-19 is mentioned from time to time. I am not sure it is necessary. I find the use of the pandemic superfluous.

What is the manuscript’s contribution to state of the art? The authors argue that they developed guidelines for mobile application accessibility, but such guidelines had already been created. Regrettably, no relevant literature review is proposed, and the results are not discussed.

Table No. 5 should not be at the end of section 3.2. Sections should not end with tables. I recommend adding some relevant text.

Author Response

First of all, we would like to thank the Reviewer for his/her time dedicated to carefully reading this manuscript. Especially in this difficult situation, we feel the need to express our gratitude for all the effort made.  Thank you very much. 

One of the main general concerns of this Reviewer is the fact that "the manuscript seems to lack features of an informatics paper". We agree that the paper is not an engineer/computer science paper but rather lies in the area of communication, social informatics and media and social sciences. We do hope that this Reviewer can regard this manuscript as such. We have submitted to this journal as there are a large number of papers published in this journal in these areas.

We just aim to study the accessibility issues raised by the appearance of apps of local press in Portugal from a social point of view, i.e., whether the local information in these apps can be easily accessed by a large number of Portuguese users. We argue that this is a relevant social issue to investigate but we do not study economic issues or the motivation of readers to install these apps. We have incorporated a lot of changes according to the issues raised in the report in order to improve the paper. We next describe these changes:

We have modified the abstract to be more concrete about the main goal and the structure of the paper. 

We have underlined the main goal of our work in the introduction and abstract, as suggested. Please see also the first part of the section "Materials and Methods".

We have added several references (see [5,26]) to complete the relatively scarce literature about the topic of this work. 

In this revised version we have added an explicit mention to the "Web Content Accessibility Guidelines 2.1" which was previously referred to in ref [19] (now [20] in the new version).

We have also clarified some methodological questions raised in the report, such as who tested the apps and collected the data (please see paragraph before Section 21.). 

As for the confusion of where the discussion is located in the manuscript, we have made this clear at the beginning of section "Results and Discussion". The section is divided in two subsections and we present the results obtained from our analysis of the apps together with a discussion of their importance in the Portuguese context. 

The mentions to the Covid-19 have been erased as suggested by the Reviewer. 

We have followed the recommendation of the Reviewer and avoided concluding the (sub)sections with a table (tables 4 and 5). 

We hope that the changes we made address satisfactorily the issues raised by the Reviewer.

Reviewer 2 Report

Section 1.2. Accessibility of mobile applications should include a mention to current EU legislation. See:

https://ec.europa.eu/growth/single-market/european-standards/harmonised-standards/accessibility-websites-and-mobile-applications_en

Missing background information about how the Indicators for the analysis of the design of Portuguese apps were selected. 

Author Response

We thank this Reviewer very much for the comments and suggestions. We have incorporated all of them. We have included the reference requested in Section 1.2, reference [19]. Further, we have extended in Section "Materials and Methods" the discussion about the indicators, explaining that they were derived by identifying the key factors of the theoretical framework developed in previous sections and adapting the indicators presented in the papers [29, 30]. The more standard indicators were directly derived from [29] and the ones devoted to elderly population were developed using [30]. We do hope that the changes we made address satisfactorily the issues raised by the Reviewer.

Round 2

Reviewer 1 Report

The authors made an attempt to develop guidelines for assessing the quality of mobile applications (including accesability). The authors used the mobile application and observed its behavior. The proposed evaluation model has its drawbacks. The app is evaluated by the so-called cognitive walkthrough. The auditor evaluates the selected attributes of the mobile app and awards a point or does not award it. The assessment is therefore based on the so-called checklist. The authors proposed a checklist for assessing the quality of mobile applications. However, the assessment is subjective. Two auditors may give different grades. The article presents a certain view (voice in the discussion), therefore its publication may be considered.

  1. The corrections introduced by the authors are of a cosmetic nature. There is still some uncertainty in the article. What is the purpose of the work (?) - assessment of accesability or development of guidelines for the assessment of accesability, and maybe both? So is the title appropriate? It is worth considering this.
  2. I suggest strengthening the section on Materials and Methods. In my opinion, the Materials and Methods chapter is weak. It is worth adding here that the authors proposed the so-called checklist and expert accesability audit performed during the so-called cognitive walkthrough.

Please see:

1) Jadhav, D., Bhutkar, G., Mehta, V. (2013, September). Usability evaluation of messenger applications for Android phones using cognitive walkthrough. In Proceedings of the 11th Asia Pacific Conference on Computer Human Interaction (pp. 9-18).

2) Mahatody, T., Sagar, M., Kolski, C. (2010). State of the art on the cognitive walkthrough method, its variants and evolutions. Intl. Journal of Human – Computer Interaction, 26 (8), 741-785.

After making these corrections, you may consider publishing an article.

Author Response

Our most heartfelt thanks to the Reviewer for his/her careful reading of the manuscript and for the useful remarks.

We have made the required changes to the manuscript (underlined in red text), namely, clarifying the purposes of this work (see abstract and introduction) and strengthening the section on Materials and Methods (adding further explanations and motivations to the use of the cognitive walkthrough methodology). Indeed we believe that this work lies in the area of social informatics more than a purely informatics paper. We do not aim to provide novel insights in the accessibility assessment area but rather analyse accessibility from a social point of view within the Portuguese context. We appreciate that the reviewer can see value in this work as a social informatics paper (a core area of this journal). We hope that the changes we made address satisfactorily the issues raised in the review. We appreciate very much the suggestions made that led to a much better paper. We acknowledged that in the section Acknowledgements.